# Polysulfurating reagent design for unsymmetrical polysulfide construction

Xiao Xiao[1], Jiahui Xue[1] & Xuefeng Jiang [1,2,3]

From life science to material science, to pharmaceutical industry, and to food chemistry, polysulfides are vital structural scaffolds. However, there are limited synthetic methods for unsymmetrical polysulfides. Conventional strategies entail two pre-sulfurated cross-coupling substrates, R–S, with higher chances of side reactions due to the characteristic of sulfur. Herein, a library of broad-spectrum polysulfurating reagents, R–S–S–OMe, are designed and scalably synthesized, to which the R–S–S source can be directly introduced for late-stage modifications of biomolecules, natural products, and pharmaceuticals. Based on the hard and soft acids and bases principle, selective activation of sulfur-oxygen bond has been accomplished via utilizing proton and boride for efficient unsymmetrical polysulfuration. These polysulfurating reagents are highlighted with their outstanding multifunctional gram-scale transformations with various nucleophiles under mild conditions. A diversity of polysulfurated biomolecules, such as SS−(+)-δ-tocopherol, SS-sulfanilamide, SS-saccharides, SS-amino acids, and SSS-oligopeptides have been established for drug discovery and development.

[1] Shanghai Key Laboratory of Green Chemistry and Chemical Process, Department of Chemistry, East China Normal University, 3663 North Zhongshan Road, Shanghai 200062, China. [2] State Key Laboratory of Elemento-Organic Chemistry, Nankai University, Tianjin 300071, China. [3] State Key Laboratory of Organometallic Chemistry, Shanghai Institute of Organic Chemistry, Chinese Academy of Sciences, 345 Lingling Road, Shanghai 200032, China. Correspondence and requests for materials should be addressed to X.J. (email: xfjiang@chem.ecnu.edu.cn)

Disulfide scaffolds, containing two covalently linked sulfur atoms, are important molecular motifs in life science[1–6], pharmaceutical science[7–15], and food chemistry[16–18] by virtue of their unique pharmacological and physiochemical properties (Fig. 1a). Disulfide bonds, for instance, in biomolecules take multifaceted roles in various biochemical redox processes to generate and regulate hormones, enzymes, growth factors, toxins, and immunoglobulins for very homeostasis and bio-signaling (e.g., metal trafficking); secondary and tertiary structures of proteins are also well formed and stabilized via the disulfide bridge[2–5]. In recent decades, potent bioactive natural products and pharmaceuticals possessing sulfur–sulfur bonds have been discovered, such as the antifungal polycarpamine family[7], the anti-poliovirus epidithiodiketopiperazine (ETPs) family[8, 9], romidepsin[10], gliotoxin[11], and some new histone deacetylase/methyltransferase inhibitors[12], which, mechanism-wise, either sequester enzyme-cofactor zinc or generate highly reactive electrophiles to induce DNA strand scission. When it comes to antibody-drug conjugates (ADC), the disulfide bond has also been extensively utilized as a linker to deliver the active drug into the targeted cell after cleavage upon internalization of ADC[19–22]. Due to the higher intracellular concentration of free thiols (glutathione) than in the bloodstream, the sulfur–sulfur bonds can be selectively cleaved in the cytoplasm of cancer cell, thereby achieving the specified release of cytotoxic molecules. Notably, disulfide compounds in allium species plants can not only demonstrate vasorelaxation activity, but also inhibit ADP-induced platelet aggregation[16–18].

Tri-sulfides have recently received considerable attention. To cite the allium-derived diallyl trisulfide (DATS) as an example, it serves as a gasotransmitter precursor and an excellent hydrogen sulfide donor, mediating and regulating the release of hydrogen sulfide upon physiological activation (Fig. 1b)[23, 24]. From the materials perspective, organotrisulfides, such as dimethyl trisulfide (DMTS) with a theoretical capacity of 849 mAhg$^{-1}$, hold promise as high-capacity cathode materials for high-energy rechargeable lithium batteries[25]. It should also be pointed out that trisulfides do exist in bioactive natural products from marine invertebrates[7, 26–28], such as the antitumor varacins A[26] and the anti-fungus outovirin C[27].

Given the importance and predominance in pharmaceuticals and other bioactive compounds of polysulfurated structures, it is always sought-after to develop general polysulfuration protocols for synthetic purposes. Although typical methods for symmetrical disulfide preparation have been well developed[29], the construction of unsymmetrical disulfides is still a challenging transformation due to the high reactivity of S–S bond[30–40]. In general, the synthesis of unsymmetrical disulfides can be achieved via an $S_N2$ process between a thiol and a prefunctionalized thiol with leaving group[32–38]. Alternatively, one can employ either two different kinds of thiols with unavoidable formation of homocoupling byproducts[39] or two distinct symmetrical disulfides with the use of rhodium(I) by Yamaguchi group[40]. Based on our continuous research in organic sulfur chemistry[41–48], comproportionation between two distinct inorganic sulfur sources was utilized for unsymmetrical disulfides syntheses[49]. However, the strategy of aforementioned methods introduces disulfide bonds from two different kinds of sulfur-containing substrates, requiring more synthetic steps and leading to side-reactions due to both reactive thio-derivatives (Fig. 2a)[30–40, 49]. We intend to develop methodology which can introduce the RSS source with one disulfurating reagent at a later stage so as to provide great compatibility and

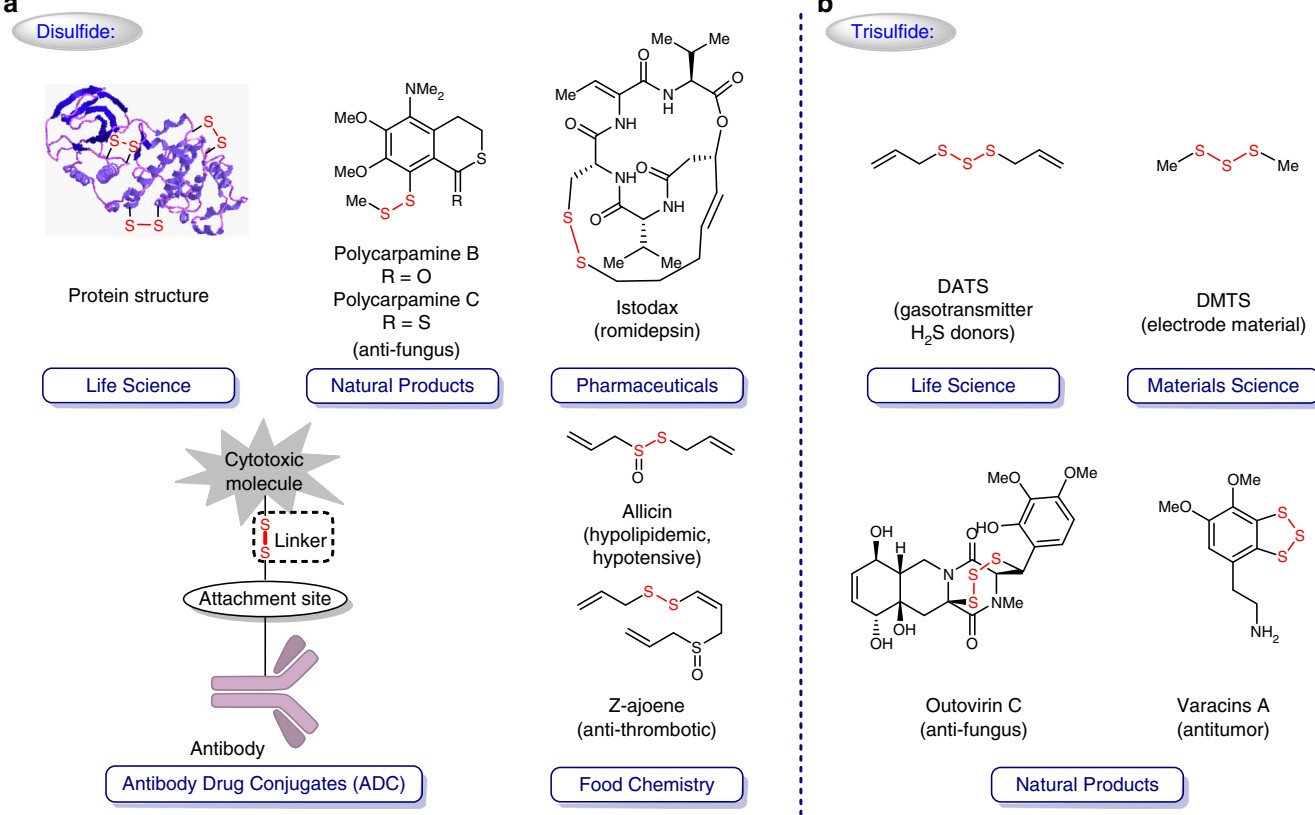

**Fig. 1** Significant polysulfides. **a** The importance of disulfide scaffolds in life science, natural products, pharmaceuticals, antibody drug conjugates, and food chemistry. **b** Functional trisulfide molecules

several possibilities of polysulfuration. Hydropersulfide (RSSH) seems to be a prime disulfurating reagent, though it is unstable owing to its high reactivity[50, 51]. Two sulfur atoms were successfully introduced in one step via oxidative cross-couplings of

acetyl masked disulfurating nucleophiles and organometallic reagents (Fig. 2b)[52].

Nevertheless, there is a large demand for a universal disulfurating reagent, which is compatible with diverse coupling

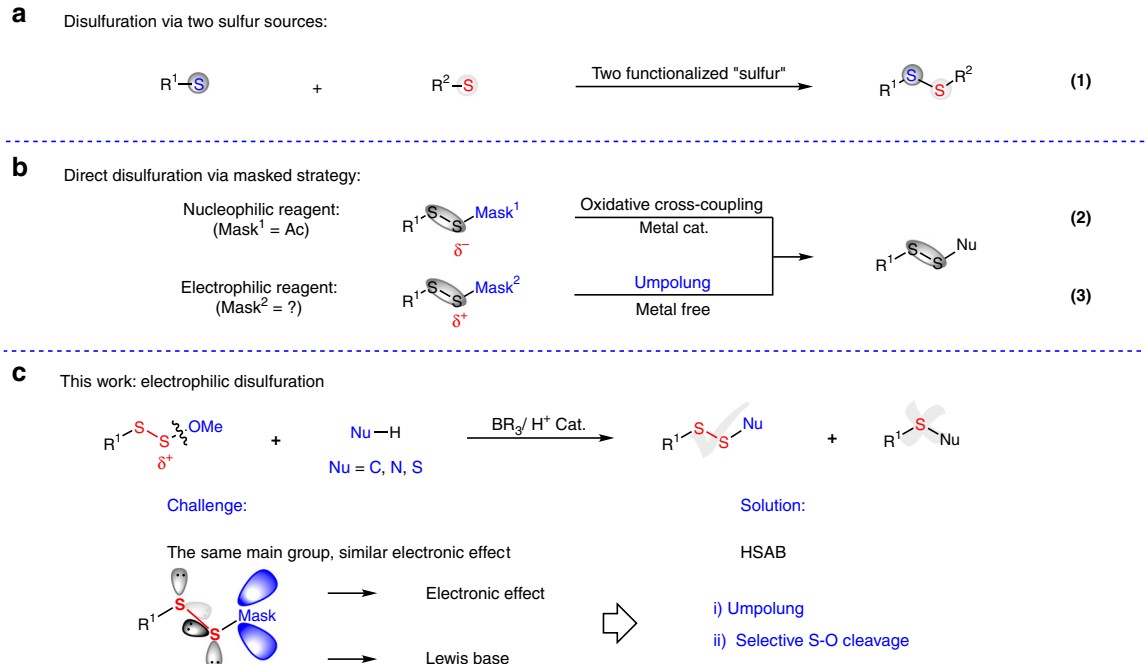

**Fig. 2** Strategies for polysulfide construction. **a** Traditional methodologies for unsymmetrical disulfide syntheses. **b** Masked strategy for disulfuration. **c** Electropilic disulfurating reagent for polysulfuration

## Table 1 Optimization of polysulfide reagents[a,b]

Conditions: 1d → 2d, CuSO₄·5H₂O, Ligand, Li₂CO₃ (1 equiv.), PhI(OPiv)₂, MeOH, temp., time

| Entry | CuSO₄ (mol%) | Ligand (mol%) | PhI(OPiv)₂ (equiv) | Temp (°C) | Time (h) | Yields (%) |
|---|---|---|---|---|---|---|
| 1[c] | 10 | bpy (10) | 2.5 | 25 | 11 | 31 |
| 2[d] | 10 | bpy (10) | 2.5 | 25 | 11 | ND |
| 3 | 10 | bpy/ phen (10) | 2.5 | 25 | 11 | 50/53 |
| 4 | 10 | L1 (10) | 2.5 | 25 | 11 | 77 |
| 5 | 10 | L2/L3/L4 (10) | 2.5 | 25 | 11 | 70/63/68 |
| 6 | 10 | L1 (10) | 2.5 | 20 | 13 | 86 |
| 7 | 5 | L1 (10) | 2.5 | 20 | 13 | 86 |
| 8 | 2.5 | L1 (10) | 2.5 | 20 | 13 | 79 |
| 9 | 5 | L1 (5) | 2.5 | 20 | 13 | 76 |
| 10 | 5 | L1 (10) | 2.2 | 20 | 13 | 88 |
| 11 | 5 | L1 (10) | 1.9 | 20 | 13 | 65 |

[Structures: L1, L2, L3, L4]

[a]Conditions: 1d (0.2 mmol, 1 equiv), CuSO₄·5H₂O, Ligand, Li₂CO₃ and PhI(OPiv)₂ were added to MeOH (2 mL) at 20 °C for 13 h
[b]Isolated yields
[c]PhI(OAc)₂ was instead of PhI(OPiv)₂
[d]PhI(OTFA)₂ was instead of PhI(OPiv)₂

**Table 2 The scope of polysulfurating reagents[a,b]**

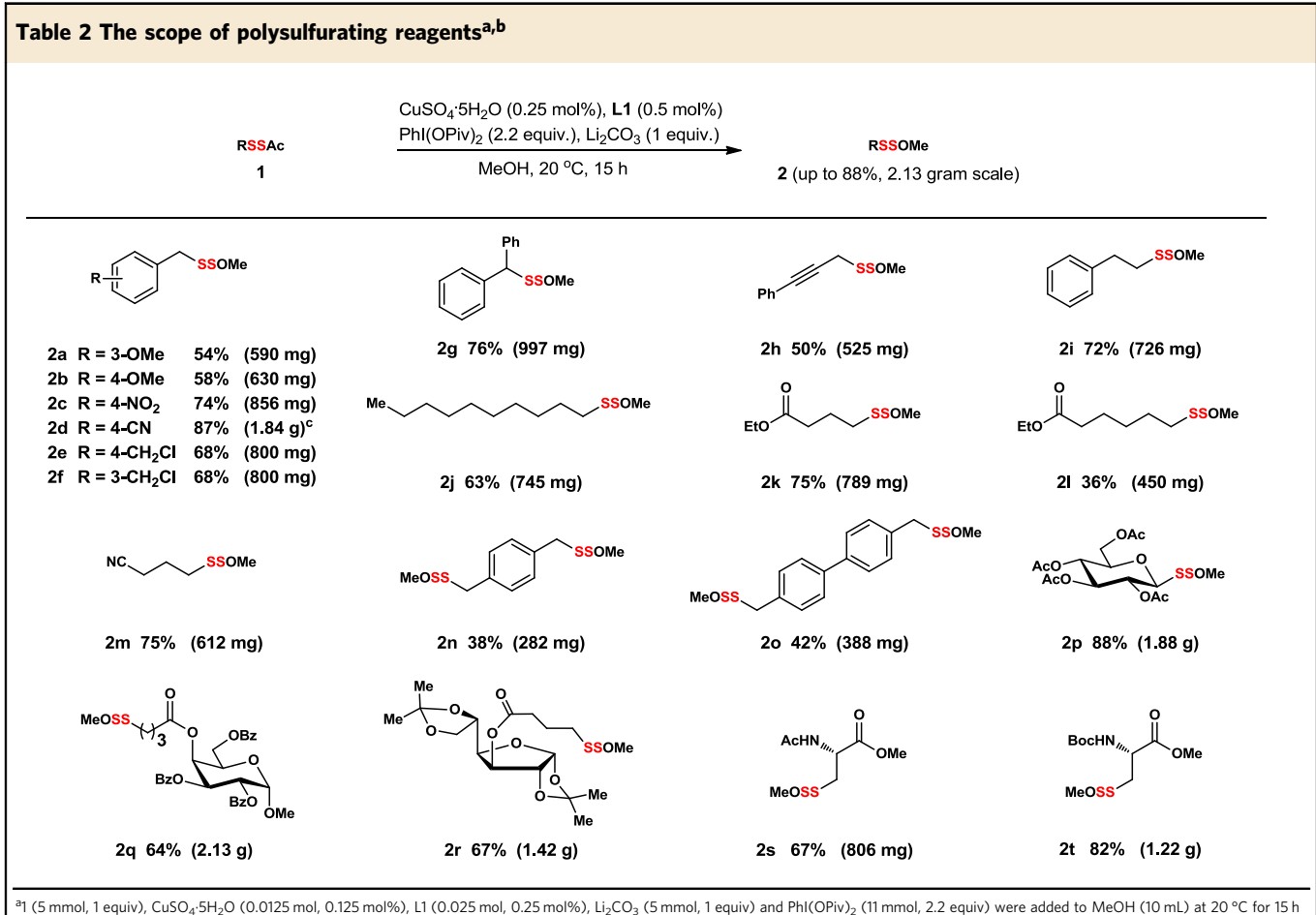

$$\text{RSSAc} \quad \xrightarrow[\text{MeOH, 20 °C, 15 h}]{\begin{array}{c}\text{CuSO}_4\cdot5\text{H}_2\text{O (0.25 mol\%), L1 (0.5 mol\%)}\\ \text{PhI(OPiv)}_2 \text{ (2.2 equiv.), Li}_2\text{CO}_3 \text{ (1 equiv.)}\end{array}} \quad \text{RSSOMe}$$

2 (up to 88%, 2.13 gram scale)

| | | |
|---|---|---|
| **2a** R = 3-OMe 54% (590 mg) | **2g** 76% (997 mg) | **2h** 50% (525 mg) |
| **2b** R = 4-OMe 58% (630 mg) | | |
| **2c** R = 4-NO₂ 74% (856 mg) | | |
| **2d** R = 4-CN 87% (1.84 g)[c] | **2j** 63% (745 mg) | **2k** 75% (789 mg) |
| **2e** R = 4-CH₂Cl 68% (800 mg) | | |
| **2f** R = 3-CH₂Cl 68% (800 mg) | | |

**2a** R = 3-OMe 54% (590 mg)
**2b** R = 4-OMe 58% (630 mg)
**2c** R = 4-NO₂ 74% (856 mg)
**2d** R = 4-CN 87% (1.84 g)[c]
**2e** R = 4-CH₂Cl 68% (800 mg)
**2f** R = 3-CH₂Cl 68% (800 mg)

**2g** 76% (997 mg)
**2h** 50% (525 mg)
**2i** 72% (726 mg)
**2j** 63% (745 mg)
**2k** 75% (789 mg)
**2l** 36% (450 mg)
**2m** 75% (612 mg)
**2n** 38% (282 mg)
**2o** 42% (388 mg)
**2p** 88% (1.88 g)
**2q** 64% (2.13 g)
**2r** 67% (1.42 g)
**2s** 67% (806 mg)
**2t** 82% (1.22 g)

[a]1 (5 mmol, 1 equiv), CuSO₄·5H₂O (0.0125 mol, 0.125 mol%), L1 (0.025 mol, 0.25 mol%), Li₂CO₃ (5 mmol, 1 equiv) and PhI(OPiv)₂ (11 mmol, 2.2 equiv) were added to MeOH (10 mL) at 20 °C for 15 h
[b]Isolated yields
[c]1 (10 mmol, 1 equiv) and MeOH (10 mL) were used

partners without transition-metal catalysis. The umpolung strategy, replacement of acetyl (RSS⁻) with methoxyl (RSS⁺) group, will afford the precursor of persulfide cation (Fig. 2c). Originating from the same main group, sulfur and oxygen possess similar electronic effect, which imposes a great challenge for selective cleavage of S–O bond with S–S bond untouched. Based on the hard and soft acids and bases (HSAB) principle[53], we hypothesize that boride/proton can help to make the difference between S–S and S–O, in which the hard acid boride/proton prefers oxygen coordination. Herein, we disclose a polysulfurating reagent which can construct unsymmetrical disulfide and trisulfide products by utilizing a RSS source only on one substrate, which renders the late-stage functionalization feasible. Different nucleophilic regents, such as 1,3-dicarbonyl derivatives, electron-rich arenes, heteroarenes, amines, and thiols, had been smoothly coupled with disulfurating reagents under mild, transition-metal-free, and base-free conditions, especially suitable for the late-stage modification of natural products and pharmaceuticals.

## Results

**Optimization and synthesis of polysulfurating reagents**. Initial studies commenced with the construction of designed electrophilic polysulfurating reagents. It was hypothesized that the electrophilic reagent could be obtained through hydropersulfide anion and methanol via oxidative cross-coupling. The polysulfurating reagent **2d** was obtained in 31% yield under the conditions of copper(II) as catalyst, 2,2′-bipyridine as ligand, and

PhI(OAc)₂ as oxidant (Table 1, entry 1). The bulky iodonium salt PhI(OPiv)₂ was the oxidant of choice in this conversion (Table 1, entries 1–3). Systematic investigations of ligands showed that 4,7-diphenyl-1,10-phenanthroline helped to increase the yield of **2d** to 77% (Table 1, entries 3–5). Further study demonstrated that slightly lower temperature was important for keeping product **2d** stable in this system (Table 1, entry 6). Catalyst loading was lowered with the same efficiency of the transformation (Table 1, entries 7–9). The optimal conditions were found to involve treatment of **1d** with 5 mol% of catalyst, 10 mol% of ligand **L1**, 2.2 equivalents of bis(tert-butylcarbonyloxy)iodobenzene, and 1.0 equivalent of lithium carbonate in 0.1 M methanol at 20 °C, which afforded electrophilic polysulfurating reagent **2d** in the yield of 88% (Table 1, entry 10). When the oxidant bis(tert-butylcarbonyloxy)iodobenzene was reduced to 1.9 equivalents, the yield of **2d** was dropped sharply to 65% (Table 1, entry 11).

With the optimized conditions in hand, the syntheses of electrophilic polysulfurating reagents were comprehensively investigated. A scale of 5 mmol operation was practicably performed, decreasing catalyst loading to 0.25 mol% (for details see the Supplementary Table 2). Various acetyl substituted disulfides were readily transformed to methoxyl substituted disulfides (Table. 2). Initially, the reagents bearing both electron-donating and electron-withdrawing groups on aromatic rings were successfully obtained (Table 2, **2a**–**2f**). Notably, 1.84 g of **2d** was achieved in a yield of 87% with 10 mmol scale operation (Table 2, **2d**). The arene substituted with

## Table 3 Disulfuration with carbon nucleophiles [a,b]

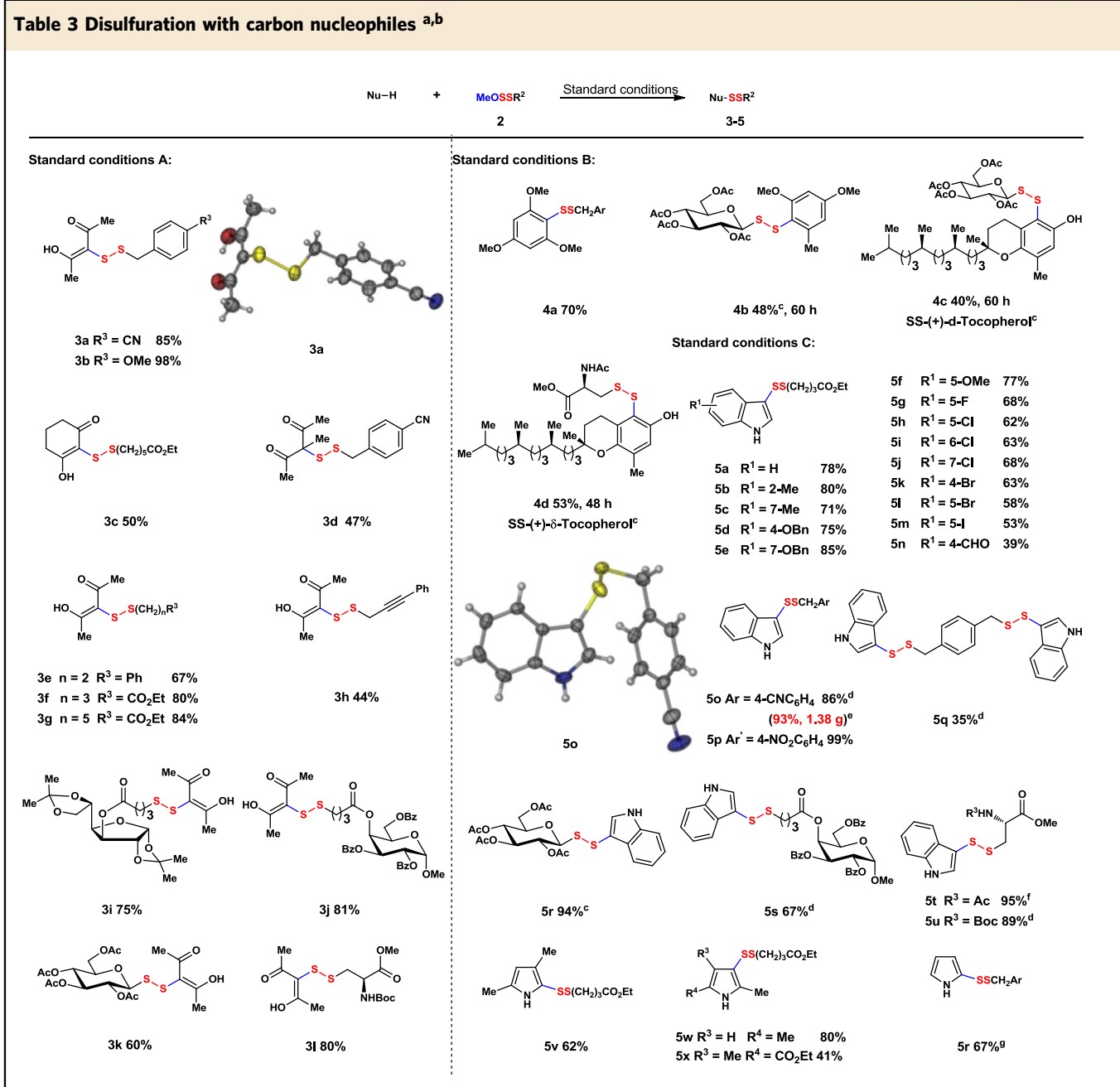

Nu−H  +  MeOSSR² ——Standard conditions——→ Nu-SSR²

**Standard conditions A:**

3a R³ = CN  85%
3b R³ = OMe  98%

3a

3c 50%

3d 47%

3e  n = 2  R³ = Ph  67%
3f  n = 3  R³ = CO₂Et  80%
3g  n = 5  R³ = CO₂Et  84%

3h 44%

3i 75%

3j 81%

3k 60%

3l 80%

**Standard conditions B:**

4a 70%

4b 48%[c], 60 h

4c 40%, 60 h
SS-(+)-d-Tocopherol[c]

**Standard conditions C:**

4d 53%, 48 h
SS-(+)-δ-Tocopherol[c]

5a R¹ = H  78%
5b R¹ = 2-Me  80%
5c R¹ = 7-Me  71%
5d R¹ = 4-OBn  75%
5e R¹ = 7-OBn  85%

5f  R¹ = 5-OMe  77%
5g  R¹ = 5-F  68%
5h  R¹ = 5-Cl  62%
5i  R¹ = 6-Cl  63%
5j  R¹ = 7-Cl  68%
5k  R¹ = 4-Br  63%
5l  R¹ = 5-Br  58%
5m  R¹ = 5-I  53%
5n  R¹ = 4-CHO  39%

5o

5o Ar = 4-CNC₆H₄ 86%[d]
(93%, 1.38 g)[e]
5p Ar' = 4-NO₂C₆H₄ 99%

5q 35%[d]

5r 94%[c]

5s 67%[d]

5t R³ = Ac  95%[f]
5u R³ = Boc  89%[d]

5v 62%

5w R³ = H  R⁴ = Me  80%
5x R³ = Me R⁴ = CO₂Et  41%

5r 67%[g]

[a]Standard conditions A: NuH (0.22 mmol, 1.1 equiv), 2 (0.2 mmol, 1 equiv), B(C₆F₅)₃ (0.01 mmol, 5 mol%) and 4-MeOPy (0.01 mmol, 5 mol%) were added to DCE (0.25 mL) at r.t. for 22 h. Standard conditions B: NuH (0.3 mmol, 1.5 equiv), 2 (0.2 mmol, 1 equiv) and B(C₆F₅)₃ (0.01 mmol, 5 mol%) were added to PhMe (0.5 mL) at 0 °C for 24 h. Standard conditions C: NuH (0.3 mmol, 1.5 equiv), 2 (0.2 mmol, 1 equiv) and MeSO₃H (0.02 mmol, 10 mol%) were added to ᵗAmylOH (0.5 mL) at 0 °C for 5–24 h
[b]Isolated yields
[c]r.t. was instead of 0 °C
[d]B(C₆F₅)₃ (0.002 mmol, 1 mol%) was used
[e]B(C₆F₅)₃ (0.01 mmol, 0.2 mol%) was used
[f]B(C₆F₅)₃ (0.004 mmol, 2 mol%) were added to PhMe (0.25 mL) at r.t. for 24 h
[g]NuH (0.22 mmol, 1.1 equiv), 2 (0.2 mmol, 1 equiv) and B(C₆F₅)₃ (0.004 mmol, 2 mol%) were added to PhMe (0.25 mL) at 0 °C for 24 h. Ar = 4-CNC₆H₄

chloromethylene group was compatible under the standard conditions (Table 2, **2e–2f**). Reactions involving secondary benzyl and propargyl derivatives were carried out smoothly (Table 2, **2g–2h**). When aliphatic substrates were evaluated, the corresponding products were formed efficiently (Table 2, **2i–2m**). The scope was further demonstrated through the successful syntheses of bis-disulfurating reagents (Table 2, **2n–2o**). Notably, the modification of saccharides and amino acids were also converted into corresponding disulfurating reagents (Table 2, **2p–2t**). These

reagents are fairly stable without deterioration when stored in a refrigerator (−18 °C) for half a year. Around 20% of these reagents will decompose at room temperature (+25 °C) after 1 week.

**Polysulfuration with designed reagents**. With the class of disulfurating reagents in hand, the construction of unsymmetrical disulfides and trisulfides was consequently explored. We initiated

## Table 4 Disulfuration with heteroatomic nucleophiles [a,b]

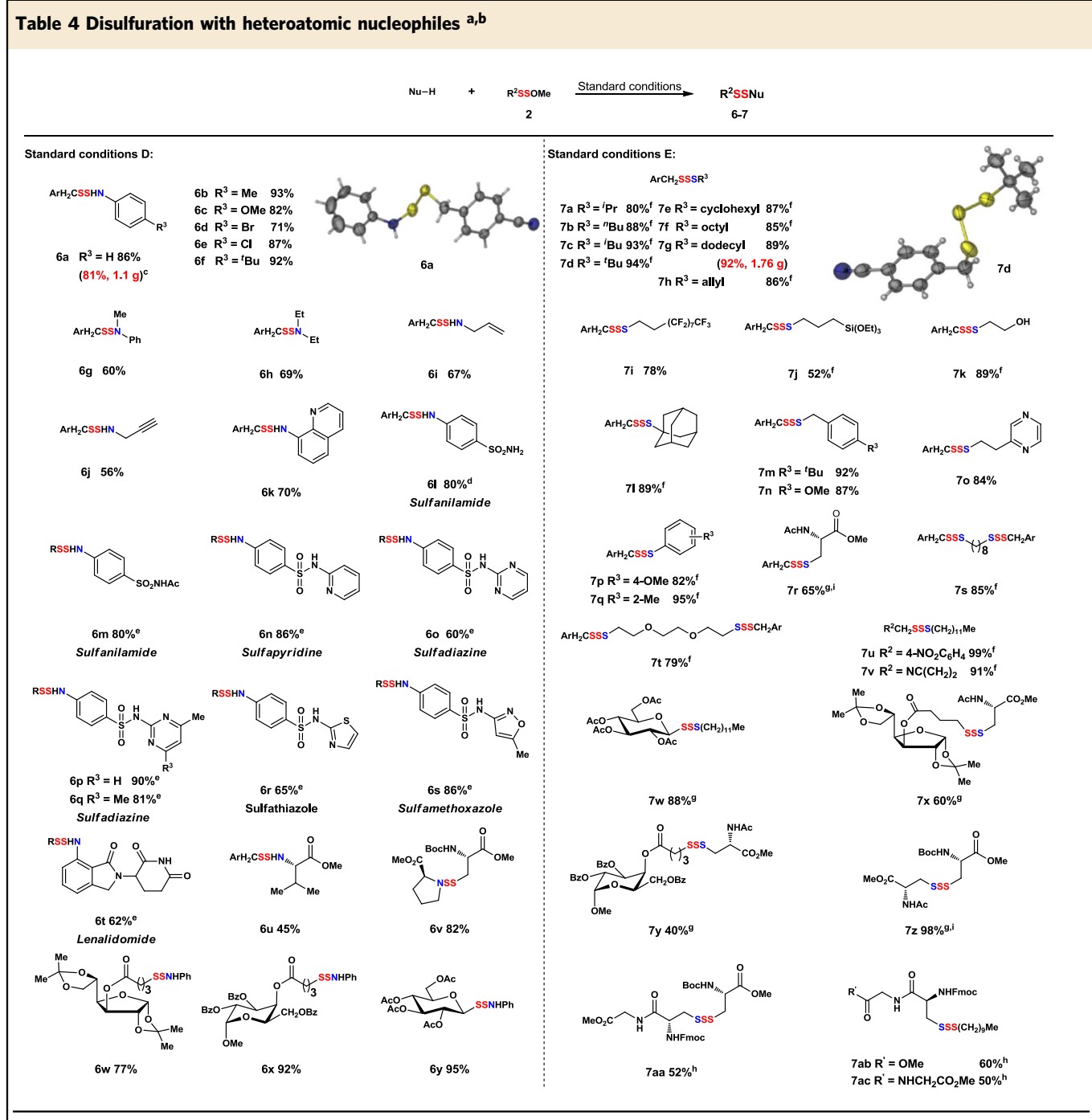

$$Nu-H \quad + \quad R^2SSOMe \xrightarrow{\text{Standard conditions}} R^2SSNu$$

2 → 6–7

**Standard conditions D:**

6a R³ = H 86% (81%, 1.1 g)[c]
6b R³ = Me 93%
6c R³ = OMe 82%
6d R³ = Br 71%
6e R³ = Cl 87%
6f R³ = ᵗBu 92%
6a
6g 60%
6h 69%
6i 67%
6j 56%
6k 70%
6l 80%[d] *Sulfanilamide*
6m 80%[e] *Sulfanilamide*
6n 86%[e] *Sulfapyridine*
6o 60%[e] *Sulfadiazine*
6p R³ = H 90%[e]
6q R³ = Me 81%[e] *Sulfadiazine*
6r 65%[e] Sulfathiazole
6s 86%[e] *Sulfamethoxazole*
6t 62%[e] *Lenalidomide*
6u 45%
6v 82%
6w 77%
6x 92%
6y 95%

**Standard conditions E:**

ArCH₂SSSR³

7a R³ = ⁱPr 80%[f]
7b R³ = ⁿBu 88%[f]
7c R³ = ⁱBu 93%[f]
7d R³ = ᵗBu 94%[f] (92%, 1.76 g)
7e R³ = cyclohexyl 87%[f]
7f R³ = octyl 85%[f]
7g R³ = dodecyl 89%
7h R³ = allyl 86%[f]
7d
7i 78%
7j 52%[f]
7k 89%[f]
7l 89%[f]
7m R³ = ᵗBu 92%
7n R³ = OMe 87%
7o 84%
7p R³ = 4-OMe 82%[f]
7q R³ = 2-Me 95%[f]
7r 65%[g,i]
7s 85%[f]
7t 79%[f]
7u R² = 4-NO₂C₆H₄ 99%[f]
7v R² = NC(CH₂)₂ 91%[f]
7w 88%[g]
7x 60%[g]
7y 40%[g]
7z 98%[g,i]
7aa 52%[h]
7ab R' = OMe 60%[h]
7ac R' = NHCH₂CO₂Me 50%[h]

[a]Standard conditions D: NuH (0.22 mmol, 1.1 equiv), 2 (0.2 mmol, 1 equiv) and B(C₆F₅)₃ (0.005 mmol, 2.5 mol%) were added to PhMe (0.5 mL) at r.t. for 24 h. Standard conditions E: NuH (0.22 mmol, 1.1 equiv) and 2 (0.2 mmol, 1 equiv) were added to DCM (2.0 mL) at r.t. for 8 h
[b]Isolated yields
[c]B(C₆F₅)₃ (0.0125 mmol, 0.25 mol%) was used
[d]CH₃CN was used as solvent
[e]NuH (0.2 mmol, 1 equiv), 2 (0.3 mmol, 1.5 equiv) and B(C₆F₅)₃ (0.005 mmol, 2.5 mol%) were added to DMF at r.t. for 24 h
[f]B(C₆F₅)₃ (2.5 mol%) was added at r.t. for 5 h
[g]B(C₆F₅)₃ (2.5 mol%) and DCM (0.5 mL) was added
[h]B(C₆F₅)₃ (2.5 mol%) and DMF (0.5 mL) was added
[i]24 h. Ar = 4-CNC₆H₄, R = (CH₂)₉Me

our efforts with 1,3-dicarbonyl compounds due to their excellent nucleophilic property. Based on the HSAB principle, the coupling between acetylacetone and reagent **2d** has been explored under the assistance of the hard acid Tris(perfluorophenyl)borane as a catalyst (for details see the Supplementary Table 3). Various 1,3-

dicarbonyl structures effectively afford disulfuration catalyzed with the combination of tris(perfluorophenyl)borane and 4-methoxypyridine (Table 3). Acyclic and cyclic 1,3-dicarbonyl substrates were smoothly converted to the desired disulfides (Table 3, **3a–3d**). The configuration of **3a** was further confirmed

through X-ray crystallographic analysis. Aliphatic and propargyl derivatives were compatible in this process (Table 3, **3e–3h**). Significantly, disulfurating reagents bearing both saccharide and amino acid groups accomplished this transformation efficiently with two parts connected via the disulfur linkage (Table 3, **3i–3l**).

Following the activation mode, electron-rich aromatics were readily accommodated under standard conditions (Table 3, **4a–4d**). (+)-δ-Tocopherol, a significant bioactive molecule, could be disulfurated directly despite the presence of free hydroxyl group (Table 3, **4c–4d**). Indole and pyrrole, ubiquitous in natural products and pharmaceuticals, are excellent coupling partners as well. Indoles bearing both electron-rich and -deficient functional groups proceeded smoothly with disulfurating reagents to afford the corresponding indolyl-disulfides on 3-position (Table 3, **5a–5p**). A bis-disulfurating electrophile also afforded the corresponding twofold disulfur-containing molecule efficiently (Table 3, **5q**). Saccharide and amino acid structures were directly installed with indoles via the disulfide linker (Table 3, **5r-5u**). A gram-scale operation was performed with 5 mmol of **2d** under the catalysis of 1 mol% of $B(C_6F_5)_3$ affording **5o** in 93% yield (1.38 g), which structure was further confirmed through X-ray analysis. In particular, iodo- and formyl-substituted indoles were also compatible in this transformation (Table 3, **5m-5n**). Pyrroles substituted on different positions were treated to the disulfuration conditions, successfully providing desired products as well (Table 3, **5v-5y**).

Subsequently, amine partners were systematically varied providing access to a wide range of functional aza-disulfide in the presence of 2.5 mol% of tris(perfluorophenyl)borane. The anilines substituted with electron-withdrawing and electron-donating functional groups afforded the desired aza-disulfides in moderate to excellent yields (Table 4, **6a-6f**). The secondary amines proceeded in this transformation, affording corresponding products in favorable yields (Table 4, **6g-6h**). Notably, allyl, propargyl and heteroaromatic amines were all efficiently transformed to the corresponding products (Table 4, **6i-6k**). Sulfanilamides, as a significant type of antibiotic, could be modified with the designed persulfurating reagent in good to excellent yields (Table 4, **6m-6s**). Lenalidomide, a myeloma drug, was installed with the disulfide under mild reaction conditions (Table 4, **6t**). Furthermore, functional disulfurating electrophiles, modified with saccharide and amino acid groups, were furnished with the substituted disulfur amine linker (Table 4, **6u-6y**). The structure of **6a** was further confirmed by X-ray analysis. In order to validate the efficiency and practicability of this aza-disulfuration, 0.25 mol% catalyst loading was launched on a gram-scale reaction to afford **6a** in 81% yield (1.1 g).

Trisulfuration was readily achieved with thiols as a nucleophile (Table 4, **7a-7q**). Even sterically bulky aliphatic thiols, *tert*-butylthiol and 1-adamantanethiol, displayed excellent trisulfurations (Table 4, **7d**, and **7l**). The structure of **7d** was further confirmed via X-ray analysis. A gram-scale production for **7g** could be performed in 92% yield practically. Thiols substituted with vinyl, polyfluoroalkyl, silyl, and hydroxyl groups, and heterocycles were all tolerated in this transformation, being converted to the unsymmetrical trisulfides, respectively (Table 4, **7h-7k**, and **7o**). Even dithiols efficiently formed the corresponding twofold trisulfur-containing products in good yields (Table 4, **7s-7t**). Aliphatic trisulfurations could be achieved in high yields (Table 4, **7u-7v**). It should be noted that trisulfides containing saccharide and cysteine fragments were readily formed through these reagents (Table 4, **7r**, **7w-7ac**). Cysteine was successfully utilized for constructing trisulfur-containing amino acids and oligopeptides, which might provide another access for peptide drug discovery (Table 4, **7aa-7ac**).

## Discussion

In summary, a class of stable and broad-spectrum polysulfurating reagents with masked strategy has been designed and a general polysulfurating methodology has been established under mild conditions, which can directly introduce two sulfur atoms into functional molecules. The designed reagents were compatible with a considerable range of significant biomolecules, such as saccharides, amino acids, peptides and variety of heterocycles. This protocol showcases the wide utility of both carbon and nitrogen nucleophiles resulting in the functional disulfides. Furthermore, the trisulfuration provides a convenient and efficient method for sulfur-containing drug discovery. Further studies on modification of biomolecules and pharmaceuticals with these disulfurating reagents are still ongoing.

## Methods

**General methods.** See Supplementary Methods for further details.

**General procedure for syntheses of disulfurating reagents 2.** To a Schlenk tube were added RSSAc **1** (5 mmol, 1 equivalent), $CuSO_4 \cdot 5H_2O$ (0.0125 mmol, 0.25 mol %, 3.2 mg), **L1** (0.025 mmol, 0.5 mol%, 8.1 mg), $Li_2CO_3$ (5 mmol, 1 equivalent, 370 mg), $PhI(OPiv)_2$ (11 mmol, 2.2 equivalents, 4.47 g) and undried MeOH (10 mL), the mixture was stirred at 20 °C under normal conditions for 15 h. Then the mixture was quenched by saturated $NaHCO_3$ and extracted by DCM before the organic phase was concentrated under vacuum without adding silica gel. Purification by column chromatography afforded the desired product.

**General procedure for syntheses of disulfides 3.** To a Schlenk tube were added 1,3-dicarbonyl compound (0.22 mmol, 1.1 equivalents), $B(C_6F_5)_3$ (0.01 mmol, 5 mol%, 5.2 mg), 4-MeO-pyridine (0.01 mmol, 5 mol%, 1.1 mg), RSSOMe **2** (0.2 mmol, 1 equivalent), and 1,2-dichloroethane (0.25 mL), the mixture was stirred at r.t. for 22 h before it was concentrated under vacuum. Purification by column chromatography afforded the desired product.

**General procedure for syntheses of disulfides 4.** To a Schlenk tube were added arene (0.3 mmol, 1.5 equivalents), $B(C_6F_5)_3$ (0.01 mmol, 5 mol%, 5.2 mg), RSSOMe **2** (0.2 mmol, 1 equivalent), and toluene (0.5 mL), the mixture was stirred at 0 °C or r.t. for 24–60 h before it was concentrated under vacuum. Purification by column chromatography afforded the desired product.

**General procedure for syntheses of disulfides 5.** Method A: To a Schlenk tube were added indole (0.3 mmol, 1.5 equivalents), $MeSO_3H$ (0.02 mmol, 10 mol%, 2 mg), RSSOMe **2** (0.2 mmol, 1 equivalent), and *t*-AmylOH (0.5 mL), the mixture was stirred at r.t. for 24 h before it was concentrated under vacuum. Purification by column chromatography afforded the desired product. Method B: To a Schlenk tube were added indole (0.22 mmol, 1.1 equivalents), $B(C_6F_5)_3$ (0.004 mmol, 2 mol %, 2.1 mg), RSSOMe (0.2 mmol, 1 equivalent), and toluene (0.25 mL), the mixture was stirred at 0 °C or r.t. for 24 h before it was concentrated under vacuum. Purification by column chromatography afforded the desired product.

**General procedure for syntheses of aza-disulfides 6.** To a Schlenk tube were added amine (0.22 mmol, 1.1 equivalents), $B(C_6F_5)_3$ (0.01 mmol, 2.5 mol%, 2.6 mg), RSSOMe **2** (0.2 mmol, 1 equivalent), and toluene (0.5 mL), the mixture was stirred at 0 °C or r.t. for 24 h before it was concentrated under vacuum. Purification by column chromatography afforded the desired product.

**General procedure for syntheses of trisulfides 7.** To a Schlenk tube were added thiol (0.22 mmol, 1.1 equivalents), $B(C_6F_5)_3$, RSSOMe **2** (0.2 mmol, 1 equivalent), and DCM (0.5 mL), the mixture was stirred at r.t. under $N_2$ atmosphere for 5–8 h before it was concentrated under vacuum. Purification by column chromatography afforded the desired product.

**Data availability.** The X-ray crystallographic coordinates for structures reported in this study have been deposited at the Cambridge Crystallographic Data Centre (CCDC), under deposition number CCDC 1565934 (**3a**), 1565935(**5o**), 1565936 (**6a**) and 1565937 (**7d**). These data can be obtained free of charge from The Cambridge Crystallographic Data Centre via www.ccdc.cam.ac.uk/data_request/cif. The authors declare that all other data supporting the findings of this study are available within the article and Supplementary Information files, and also are available from the corresponding author on reasonable request.

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

## Acknowledgements

The authors are grateful for financial support provided by The National Key Research and Development Program of China (2017YFD0200500), NSFC (21722202, 21672069, 21472050), Fok Ying Tung Education Foundation (141011), DFMEC (20130076110023), the program for Shanghai Rising Star (15QA1401800), Professor of Special Appointment

(Eastern Scholar) at Shanghai Institutions of Higher Learning, and National Program for Support of Top-Notch Young Professionals.

## Author contributions

X.J. conceived the idea and supervised the whole project. X.X. designed and carried out the experiments. J.X. contributed to part experiments. X.J. and X.X. discussed the results, contributed to writing the manuscript, and commented on the manuscript. All authors approved the final version of the manuscript for submission.

## Additional information

**Competing interests:** The authors declare no competing interests.

