## [Peer Review File · Nature Communications]

Reviewer #1 (Remarks to the Author):

While there is some very interesting and useful organosulfur chemistry presented in this manuscript, unfortunately it is very poorly presented and is unacceptable for publication in its present form. In addition to numerous errors of grammar (e.g., frequent omission of articles "the" and "a") and style, the bombastic and overstated language is inappropriate for a scientific paper. For example, I have never seen words such as "triumphantly", "famous", "paramount", "preeminently", "prosperously", "excitingly", or "remarkably" used in typical chemistry journal articles. These are gross exaggerations. Even phrases such as "enormous challenges" and "burgeoning and urgent demand" are overblown. The use of slang expressions such as "Besides," "What's more" and "agreeably compatible" also makes the paper appear unprofessionally written. The authors would do well to follow the style and language used in typical mainstream organic chemistry journals and allow the interesting chemistry to sell itself rather than annoy the reader with ill-chosen and over-blown language. If the authors extensively revise the language so that it simply describes the new chemistry discovered with appropriate background and examples of applications, then the paper might well be worthy of publication in Nature Communications. For the author's benefit, I have highlighted in yellow most of the offending or ungrammatical words or phrases. I also question the description of the S-S bond as being weak -- at 70 kcal/mol it is not at all weak (for example, compared to an O-O bond with 35 kcal/mol BDE), but it is quite reactive with radicals and nucleophiles, e.g., leading to ready formation of mixtures/scrambling from unsymmetrical disulfides. The author should be careful here to distinguish between thermodynamic stability versus kinetic reactivity -- relatively strong bonds can still be quite reactive.

Reviewer #2 (Remarks to the Author):

The authors have prepared a series of polysulfurating agents by reaction of acetyl derivatives of disulfides with 4,7-diphenyl-1,10-phenanthroline, 2.2 equivalents of bis (tert-butylcarbonyloxy)iodobenzene, and 1.0 equivalent of lithium carbonate in methanol. These reagents were then used for reaction with a series of nucleophiles like arenes, heteroarenes, 1, 3-dicarbonyls, amines, and thiols to form the corresponding unsymmetrical disulfides in good yields. The method was then extended to the synthesis of trisulfide derivatives as well. The method developed appears to be good for polysulfuration and indeed it would be useful. However, I am not sure whether it is of such importance that it needs to be published in Nature Communication.

The manuscript has been drafted very poorly. There are quite a few grammatical errors. Arguments in the manuscript in support of a communication in Nature are not convincing.

Additionally, there are a number of efficient methods reported in the literature for the formation of unsymmetrical disulfides. A few of them are given below.

- 1.(Use of 1-chlorobenzotriazole ---R. Hunter, M. Caira, N. Stellenboom, J. Org. Chem., 2006, 71, 8268-8271.;
- 2.(Use of bis-(5,5-dimethyl-2-thiono-1,3,2-dioxaphosphorinanyl)disulfide - S. Antoniow, D. Witt, Synthesis, 2007, 363-366.)
- 3.(organophosphorus sulfonyl bromide as activating agent -- Synthesis, 2007, 3528-3534.)
- 4.(N-trifluoroacetyl arenesulfenamides with thiols and amines-- M. Bao, M. Shimizu, Tetrahedron, 2003, 59, 9655-9659.)

Many of these methods can be used to access the compounds reported in this communication. The manuscript is good for publication in a more specialized journal.

Reviewer #3 (Remarks to the Author):

This article is described about copper-catalyzed synthesis of RSSOMe from RSSAc and its application. Authors have been recently reported that the preparation of RSSAc from ToISO_2SNa and its reactivity in *Angew. Chem. Int. Ed.* However, the paper reveals that RSSOMe can use to widespread substrates. The procedure can produce various unsymmetrical polysulfides such as disulfides or trisulfides by using nucleophilic reagents. These obtained products is very interesting. The research shows valuable results. I'm sure that the article is suitable for the journal. After minor revision, I would like to recommend for the publication.

- 1.Is the RSSOMe stable? It should be described about its stability in the text or the reference.
- 2.Page 6, table 1: In the conversion of RSSAc into RSSOMe, Is the formation of sulfinic esters or other compounds observed?
- 3.Page 6, table 1: Was other oxidants such as O_2 or metal salts investigated?
- 4.Reference: Recently, the preparation of unsymmetrical disulfides and the copper-catalyzed sulfenamides using thiosulfonates have been reported: *Tetrahedron* 2017, 73, 2030.

Responds to the reviewers' comments:

Reviewer 1:

Q1: While there is some very interesting and useful organosulfur chemistry presented in this manuscript, unfortunately it is very poorly presented and is unacceptable for publication in its present form.

A1: Thanks for your suggestion. The presentation of the whole manuscript has been carefully polished in the revised version according to your recommendation.

Q2: In addition to numerous errors of grammar (e.g., frequent omission of articles "the" and "a") and style, the bombastic and overstated language is inappropriate for a scientific paper. For example, I have never seen words such as "triumphantly", "famous", "paramount", "preeminently," "prosperously," "excitingly", or "remarkably" used in typical chemistry journal articles. These are gross exaggerations. Even phrases such as "enormous challenges" and "burgeoning and urgent demand" are overblown. The use of slang expressions such as "Besides," "What's more" and "agreeably compatible" also makes the paper appear unprofessionally written. The authors would do well to follow the style and language used in typical mainstream organic chemistry journals and allow the interesting chemistry to sell itself rather than annoy the reader with ill-chosen and over-blown language.

A2: Thanks for your suggestion. The errors of grammar have been carefully checked and corrected in the revised manuscript. The unsuitable language has been corrected or deleted. Based on your advice, all parts of

the manuscript have been revised and polished with the help of native speakers. Please see details in the revised manuscript.

The others have been revised and the details are as follows:

1. The sentence of “Initial studies were commenced with the construction of designed electrophilic polysulfurating reagents.” has been deleted the word of “were” and has been revised as “**Initial studies commenced with the construction of designed electrophilic polysulfurating reagents.**”
2. The sentence of “It was hypothesized that the electrophilic reagent could be achieved through hydropersulfide anion and methanol under the oxidative cross-coupling.” has been revised as “It was hypothesized that the electrophilic reagent **could be obtained** through hydropersulfide anion and methanol via oxidative cross-coupling.”
3. The word of “Fortunately” has been deleted.
4. The sentence of “The bulky iodonium salt of $\text{PhI}(\text{OPiv})_2$ was more efficient oxidant in this conversion (Table 1, entries 1-3).” has been revised as “The bulky iodonium salt of $\text{PhI}(\text{OPiv})_2$ was **the optimized oxidant** in this conversion (Table 1, entries 1-3).”
5. The word of “enhance” has been corrected as “increase”.
6. The word of “slightly lowering temperature” has been revised as “slightly lower temperature”.
7. The words of “solution of methanol” have been revised as “methanol”.
8. The sentence of “5 mmol scale operation was practicably performed, decreasing catalyst loading to 0.25 mol% (for details see the Supporting Information).” has been revised as “**A scale of 5 mmol operation was**

practicably performed, decreasing catalyst loading to 0.25 mol% (for details see the Supporting Information).”

9. The phrase of “What’s more” has been revised as “Notably”.

10. The words of “agreeably compatible” have been revised as “compatible”.

11. The word of “Besides” has been deleted.

12. The phrase of “What’s more” has been deleted.

13. The word of “triumphantly” has been deleted.

14. The word of “Remarkably” has been deleted.

15. The sentence of “Lenalidomide, a myeloma drug, is well equipped with disulfides under mild reaction conditions (Table 4, **6t**).” has been revised as “Lenalidomide, a myeloma drug, is **installed with** disulfides under mild reaction conditions (Table 4, **6t**).”.

16. The word of “prosperously” has been deleted.

17. The sentence of “Excitingly, even sterically bulky aliphatic thiols *tert*-butylthiol and 1-adamantanethiol displayed excellent trisulfurations (Table 4, **7d**, and **7l**).” has been revised as “Even sterically bulky aliphatic thiols, ***tert*-butylthiol and 1-adamantanethiol**, displayed excellent trisulfurations (Table 4, **7d**, and **7l**).”.

18. The sentence of “Thiols substituted with vinyl, polyfluoroalkyl, silane, hydroxyl, and heterocycle were all tolerant in this transformation, converted to the unsymmetrical trisulfides respectively (Table 4, **7h-7k**, and **7o**).” has been revised as “Thiols substituted with **vinyl, polyfluoroalkyl, silyl, and hydroxyl groups, and heterocycles** were all

tolerated in this transformation, being converted to the unsymmetrical trisulfides respectively (Table 4, **7h-7k**, and **7o**).”.

19. The word of “Remarkably” has been deleted.

20. The sentence of “What’s more, cysteine, a significant intracellular life motif, was successfully utilized for constructing trisulfur-containing amino acids and oligopeptides, which may provide a new access for peptide drug discovery (Table 4, **7aa-7ac**).” has been revised as “**Cysteine**, was successfully utilized for constructing trisulfur-containing amino acids and oligopeptides, which might provide a new access for peptide drug discovery (Table 4, **7aa-7ac**).”.

21. The sentence of “In summary, a novel class of stable and broad-spectrum polysulfurating reagents with masked strategy has been practicably designed and universal persulfuration are consequently established under mild conditions.” has been revised as “In summary, a novel class of stable and broad-spectrum polysulfurating reagents with masked strategy **has been designed and a general polysulfurating methodology has been established under mild conditions, which can directly introduce two sulfur atoms into functional molecules.**”

22. The sentence of “This protocol serves as a fascinating protocol with toleration of both carbonic and nitrogenous nucleophiles resulting in the functional disulfides.” has been revised as “**This protocol showcases wide utility of** both carbonic and nitrogenous nucleophiles resulting in the functional disulfides.”

23. The sentence of “ Furthermore, trisulfuration is an indispensable property of these unique reagents, providing a convenient and efficient method for sulfur-containing drug discovery.” has been revised as

“Furthermore, **the trisulfuration provides** a convenient and efficient method for sulfur-containing drug discovery.”

Q3: If the authors extensively revise the language so that it simply describes the new chemistry discovered with appropriate background and examples of applications, then the paper might well be worthy of publication in Nature Communications. For the author's benefit, I have highlighted in yellow most of the offending or ungrammatical words or phrases.

A3: Thanks for your patient and careful revision. Editing works you mentioned have all been done.

Q4: I also question the description of the S-S bond as being weak -- at 70 kcal/mol it is not at all weak (for example, compared to an O-O bond with 35 kcal/mol BDE), but it is quite reactive with radicals and nucleophiles, e.g., leading to ready formation of mixtures/scrambling from unsymmetrical disulfides. The author should be careful here to distinguish between thermodynamic stability versus kinetic reactivity -- relatively strong bonds can still be quite reactive.

A4: Thanks for your suggestion. The description of BDE, which may cause confusion, has been removed in the revised manuscript. “The description of the S-S bond as being weak” has been revised as “the high reactivity of S-S bond” in the manuscript.

Reviewer 2:

Q1: The manuscript has been drafted very poorly. There are quite a few grammatical errors.

A1: Thanks for your reminding. The grammatical errors have been carefully corrected and highlighted in the revised manuscript.

Q2: There are a number of efficient methods reported in the literature for the formation of unsymmetrical disulfides. A few of them are given below.

1.(Use of 1-chlorobenzotriazole ---R. Hunter, M. Caira, N. Stellenboom, J. Org. Chem., 2006, 71, 8268-8271.;

2.(Use of bis-(5,5-dimethyl-2-thiono-1,3,2-dioxaphosphorinanyl)disulfide - S. Antoniow, D. Witt, Synthesis, 2007, 363-366.)

3.(organophosphorus sulfenyl bromide as activating agent -- Synthesis, 2007, 3528-3534.)

4.(N-trifluoroacetyl arenesulfenamides with thiols and amines-- M. Bao, M. Shimizu, Tetrahedron, 2003, 59, 9655-9659.)

Many of these methods can be used to access the compounds reported in this communication.

A2: Thanks for your kind reminding. The references (please see ref. 33, 35-37) of these methods reported you mentioned have been cited. The strategy of aforementioned methods introduces disulfide bonds from two different parts of sulfur-containing substrates, requiring more synthetic steps and leading to side-reactions due to both of reactive thio-derivatives, which impedes the wide application in chemical biology and drug

discovery. By contrast, our designed reagents and method construct unsymmetrical disulfides via utilizing the R-S-S source with onefold disulfurating reagent at late-stage, which consequently provides great compatibility and several possibilities of polysulfuration. These efficiencies play an important role for synthesis application, drug discoveries and even biological orthogonality. And those details have been added in the revised manuscript.

Reviewer 3:

Q1: Is the RSSOMe stable? It should be described about its stability in the text or the reference.

A1: Thanks for your kind reminding. The stability of reagents has been described in the part of optimization and synthesis of the polysulfurating reagent as follows: “These reagents are fairly stable without deterioration when stored in fridge (-18°C) for half a year. Around 20% of them will be decomposed at room temperature (+25°C) after one week.”

Q2: Page 6, table 1: In the conversion of RSSAc into RSSOMe, is the formation of sulfinic esters or other compounds observed?

A2: Thanks for your kind reminding. In the conversion, the sulfinic esters have never been observed.

Q3: Page 6, table 1: Was other oxidants such as O₂ or metal salts investigated?

A3: Thanks for your suggestion. The oxidants of O₂ and NFSI have been investigated but without desired product detected under the conditions. All the other metal oxidants did not work in this sensitive transformation. The details are as follows:

Optimization of polysulfide reagents.^a

entry	[M]	oxidants	yields (%) ^b
1	CuSO ₄ ·5H ₂ O	PhI(OAc) ₂	31
2	CuSO ₄ ·5H ₂ O	O ₂	ND
3	CuSO ₄ ·5H ₂ O	NFSI	ND
4	PdCl ₂	PhI(OAc) ₂	ND
5	AuPPh ₃ Cl	PhI(OAc) ₂	ND
6	FeSO ₄ ·7H ₂ O	PhI(OAc) ₂	ND
7	CuCl	PhI(OAc) ₂	27
8	CuBr	PhI(OAc) ₂	ND
9	CuI	PhI(OAc) ₂	16
10	CuCl ₂	PhI(OAc) ₂	21
11	Cu(OAc) ₂	PhI(OAc) ₂	20

^a Conditions: **1d** (0.2 mmol, 1 equiv.), [M] (10 mol%), Bipy (10 mol%), Li₂CO₃ (1.0 equiv.) and PhI(OAc)₂ (2.5 equiv.) were added to MeOH (2 mL) at 25 °C for 11 h. ^b Isolated yields.

Q4: Reference: Recently, the preparation of unsymmetrical disulfides and the copper-catalyzed sulfenamides using thioslfonates have been reported: Tetrahedron 2017, 73, 2030.

A4: Thanks for your kind reminding. This related reference has been added as ref. 38 in revised manuscript.

Reviewer #1 (Remarks to the Author):

This paper contains some nice chemistry, worthy of publication in Nature Communications. While the manuscript has been somewhat improved in this revision, it is still unacceptable due to poor English. The authors are advised to engage an editor expert in English to correct the remaining numerous problems, unless Nature Communications is willing to extensively copy-edit this paper. As a Reviewer my time is limited to suggest changes. Below I identify English language problems, and make a few suggestions for corrections; I also identify some but not all of the sentences which still do not make sense. I believe that it is the author's responsibility to submit a revised manuscript in excellent English for the paper to be acceptable for publication.

Line 49, "by virtue" not "in virtues"

Line 52, meaning of "for very homeostasis and bio-signaling" is unclear; please revise

Line 58, "romidepsin and gliotoxin" should not be capitalized.

Lines 58, 59, "sequester" and "generate" -- subject is plural

Line 66, allium species plants

Line 70, the use of "on the other hand" seems out of place here because there is nothing to contrast – both disulfides and trisulfides are important and widely discussed

Line 75, delete "do"

Line 83, "a challenging"

Line 84 "synthesis of the unsymmetrical disulfide"

Line 86 "with unavoidable formation of homocoupling products"

Line 95 "at a late-stage"

Line 109 "a R–S–S source" or better, "a RSS source" [use m-dashes not hyphens for bonds]

Line 126 "helped"

Line 135 delete "of"

Line 144 "a yield" not "the yield"

Line 146 "Reactions involving secondary..."

Line 147 "the corresponding"

Line 152 "a refrigerator"; "these reagents will decompose..."

Lines 167-168 "3-methyl-acetylacetone bearing challenging steric hindrance" this makes no sense; also, there should only be a hyphen after "3"

Lines 170-171 “amino acid groups”

Line 176 “despite of free hydroxyl group”

Line 178 “partners”; “indoles”

Line 182 “stalled” doesn’t make sense in this context

Line 183 “clicked by disulfur linker” doesn’t make sense

Line 199 “yields in late-staged” doesn’t make sense

Line 207 “achieved” not “afforded”

Line 215 “which are potential hydrogen sulfide storages for slow releasing” does not make sense

Line 226 “a considerable”

Line 228 “the wide”

Reviewer #3 (Remarks to the Author):

This revised article is corrected satisfactorily. Therefore, I would like to recommend for the publication Nature communication.

Responds to the reviewers' comments:

Reviewer 1:

Q1: While the manuscript has been somewhat improved in this revision, it is still unacceptable due to poor English. The authors are advised to engage an editor expert in English to correct the remaining numerous problems, unless Nature Communications is willing to extensively copy-edit this paper.

A1: Thanks for your great patient and careful revision for our manuscript. The presentation of the whole manuscript has been carefully polished once again in the revised version with the help of native speakers. Please see details in the revised manuscript.

Q2: Line 49, “by virtue” not “in virtues”

A2: Thanks for your suggestion. The phrase “in virtues” has been revised to “by virtue” in the revised manuscript.

Q3: Line 52, meaning of “for very homeostasis and bio-signaling” is unclear; please revise

A3: Thanks for your suggestion. The phrase of “for very homeostasis and bio-signaling” has been revised to “for very homeostasis and bio-signaling (e.g. metal trafficking)” in the manuscript.

Q4: Line 58, “romidepsin and gliotoxin” should not be capitalized.

A4: Thanks for your suggestion. The mistake has been corrected in the manuscript.

Q5: Lines 58, 59, “sequester” and “generate” -- subject is plural

A5: Thanks for your kind reminding. The two words have been corrected in the manuscript.

Q6: Line 66, allium species plants

A6: It has been revised in the manuscript.

Q7: Line 70, the use of “on the other hand” seems out of place here because there is nothing to contrast – both disulfides and trisulfides are important and widely discussed

A7: The phrase of “on the other hand” has been deleted in the manuscript.

Q8: Line 75, delete “do”

A8: The word of “do” has been deleted in the manuscript.

Q9: Line 83, “a challenging”

A9: The word of “challenging” has been revised to “a challenging” in the manuscript.

Q10: Line 84 “synthesis of the unsymmetrical disulfide”

A10: The phrase of “synthesis of” has been added in the manuscript.

Q11: Line 86 “with unavoidable formation of homocoupling products”

A11: The phrase of “formation of” has been added in the manuscript.

Q12: Line 95 “at a late-stage”

A12: The phrase of “at a late-stage” has been revised to “at a later stage” in the manuscript.

Q13: Line 109 “a R–S–S source” or better, “a RSS source” [use m-dashes not hyphens for bonds]

A13: All of the same mistakes have been revised in the manuscript.

Q14: Line 126 “helped”

A14: The word of “help” has been revised to “helped” in the manuscript.

Q15: Line 135 delete “of”

A15: The word of “of” has been deleted.

Q16: Line 144 “a yield” not “the yield”

A16: The phrase of “the yield” has been revised to “a yield”.

Q17: Line 146 “Reactions involving secondary...”

A17: The phrase of “Reactions involving” has been added.

Q18: Line 147 “the corresponding”

A18: The phrase of “the corresponding” has been added.

Q19: Line 152 “a refrigerator”; “these reagents will decompose...”

A19: The mistakes have been revised in the manuscript.

Q20: Lines 167-168 “3-methyl-acetylacetone bearing challenging steric hindrance” this makes no sense; also, there should only be a hyphen after “3”

A20: Thanks for your suggestion. The sentence of “3-methyl-acetylacetone bearing challenging steric hindrance...” has been deleted in the manuscript.

Q21: Lines 170-171 “amino acid groups”

A21: The word of “groups” has been added.

Q22: Line 176 “despite of free hydroxyl group”

A22: The phrase of “despite of free hydroxyl group” has been revised to “despite free hydroxyl group”.

Q23: Line 178 “partners”; “indoles”

A23: The words of “partner” and “indole” have been revised to “partners” and “indoles”

Q24: Line 182 “stalled” doesn’t make sense in this context

A24: The word of “stalled” has been revised to “installed”.

Q25: Line 183 “clicked by disulfur linker” doesn’t make sense

A25: The phrase of “clicked by disulfur linker” has been revised to “via the disulfur linker”.

Q26: Line 199 “yields in late-staged” doesn’t make sense

A26: The phrase of “in late-staged” has been deleted.

Q27: Line 207 “achieved” not “afforded”

A27: The word of “afforded” has been revised to “achieved”.

Q28: Line 215 “which are potential hydrogen sulfide storages for slow releasing” does not make sense

A28: The sentence of “which are potential hydrogen sulfide storages for slow releasing” has been deleted.

Q29: Line 226 “a considerable”

A29: The word of “considerable” has been revised to “a considerable”.

Q30: Line 228 “the wide”

A30: The word of “wide” has been revised to “the wide”.